# 17β-estradiol Enhances 5-Fluorouracil Anti-Cancer Activities in Colon Cancer Cell Lines

**DOI:** 10.3390/medsci10040062

**Published:** 2022-10-28

**Authors:** Amani A. Mahbub

**Affiliations:** Laboratory Medicine Department, Faculty of Applied Medical Sciences, Umm Al-Qura University, P.O. Box 715, Makkah 21955, Saudi Arabia; aamahbob@uqu.edu.sa

**Keywords:** colon cancer cells, 17β-estradiol, 5-Fluorouracil, cell cycle, sub-G1 phase, apoptosis

## Abstract

Background: 5-Fluorouracil (5-FU) represents one of the major constituents of chemotherapy combination regimens in colon cancer (CRC) treatments; however, this regimen is linked with severe adverse effects and chemoresistance. Thus, developing more efficient approaches for CRC is urgently needed to overcome these problems and improve the patient survival rate. Currently, 17β-estradiol (E2) has gained greater attention in colon carcinogenesis, significantly lowering the incidence of CRC in females at reproductive age compared with age-matched males. Aims: This study measured the effects of E2 and/or 5-FU single/dual therapies on cell cycle progression and apoptosis against human HT-29 female and SW480 male primary CRC cells versus their impact on SW620 male metastatic CRC cells. Methods: The HT-29, SW480, and SW620 cells were treated with IC_50_ of E2 (10 nM) and 5-FU (50 μM), alone or combined (E+F), for 48 h before cell cycle and apoptosis analyses using flow cytometry. Results: The data here showed that E2 monotherapy has great potential to arrest the cell cycle and induce apoptosis in all the investigated colon cancer cells, with the most remarkable effects on metastatic cells (SW620). Most importantly, the dual therapy (E+F) has exerted anti-cancer activities in female (HT-29) and male (SW480) primary CRC cells by inducing apoptosis, which was preferentially provoked in the sub-G_1_ phase. However, the dual treatment showed the smallest effect in SW620 metastatic cells. Conclusion: this is the first study that demonstrated that the anti-cancer actions of 17β-estradiol and 5-Fluorouracil dual therapy were superior to the monotherapies in female and male primary CRC cells; it is proposed that this treatment strategy could be promising for the early stages of CRC. At the same time, 17β-estradiol monotherapy could be a better approach for treating the metastatic forms of the disease. Nevertheless, additional investigations are still required to determine their precise therapeutic values in CRC.

## 1. Introduction

Colon cancer (CRC) is the third most prevalent malignancy and the second leading cause of cancer-related deaths worldwide [1,2]. Dysregulation of the cell cycle progression and apoptosis induction are critical features of cancer [3,4]. In this regard, colon carcinogenesis is commonly linked with abnormal cell cycle progression, in conjunction with the inhibition of apoptosis induction [5,6,7,8,9]. Thus, the development of new anti-cancer agents primarily focuses on regulating the cell cycle and the apoptosis process and their related molecular targets in cancer therapy [3,4].

Until now, 5-Fluorouracil (5-FU) has represented one of the most common and effective used agents and the main constituent of chemotherapy combination regimens in CRC treatments [10]. Unfortunately, the 5-FU regimen is markedly associated with toxicity, severe side effects, and chemoresistance [10,11,12,13,14]. Combining 5-FU with other cytotoxic drugs, such as oxaliplatin, capecitabine, and others, can enhance the chemotherapy outcomes [15,16]. However, the efficacy of these agents is still limited as many CRC patients have been reported to have developed chemoresistance [15,16]. Recently, Moutabian et al. (2022) [14] have reported that identifying novel, effective, and safe therapeutic anti-cancer agents for their potential application when combined with 5-FU during CRC treatment is urgently required to improve patient survival and overcome the adverse effects.

Recently, the 17β-oestradiol (E2) in colon carcinogenesis has gained greater attention as many studies have seen a significantly lower incidence of CRC in females of reproductive age compared with age-matched males [17,18]. Worldwide, the incidence of CRC is lower in females than in males, suggesting a protective role of sex hormones in CRC development [1,2,19,20,21]. In the Kingdom of Saudi Arabia, the incidence of CRC is also classified as the first and the third among males and females, respectively [2,21,22,23,24]. Better survival rates and prognosis have also been detected in female CRC patients aged between 18 and 44 years than in male patients of the same age or females older than 50 years [1,19,25]. Concurrently, the development of CRC risk increased markedly by more than 20% in females after unilateral and bilateral oophorectomy [26,27]. In addition, the use of oral contraceptives (OC) or hormone replacement therapy (HRT) by postmenopausal females was linked with a remarkable reduction in the incidence of CRC compared with age-matched non-user females [17,18]. Under this theme, Labadie et al. (2020) [28] have also found a strong inverse association between HRT application and the overall CRC risk in postmenopausal females, suggesting efficient chemopreventive actions for HRT in the suppression of the adenoma–carcinoma in CRC. Most current meta-analyses, experimental studies, and reports also reveal anti-cancer activities for 17β-oestradiol against colon carcinogenesis, which could suppress cell growth and induce apoptosis [28,29,30,31,32,33,34,35,36,37,38].

Thus, understanding the role of female sex steroids such as estrogen could be necessary to consider for the CRC treatment plan and to establish the screening, prevention, and/or treatment protocols to reduce mortality and improve the survival outcome and life quality of CRC patients. Although the ovaries are the main producers of estrogens, other peripheral tissues also express the enzymes that are required for estrogen synthesis [39,40]. The most biologically active estrogen isoform is 17β-estradiol (E2), which initiates its cellular signals by activating its nuclear receptors (ERα and ERβ), which mediate major genomic actions [41,42,43]. The E2 synthesising enzymes and ERβ are expressed in normal colonic mucosa in both genders, and CRC is mainly linked with pathological alterations in the expression of these molecules [34]. Moreover, the expression of ERβ is commonly downregulated, while ERα increases substantially in malignant colonic tissues, suggesting that E2 could induce its anti-cancer actions in the colon via ERβ [44,45,46,47,48,49,50]. In vitro studies showed that E2 therapy can inhibit the expression of ERα and activate the ERβ in colon cancer cells, which can also consequently prevent the activation of pro-oncogenic molecules [51,52].

Recently, Mahbub (2022) [53] reviewed the current pre-clinical experiments and concluded that E2 exogenous treatment and/or the reactivation of its beta receptor (ERβ) can reduce cell proliferation, promote cell cycle arrest, and induce apoptosis via the modulation of several molecular pathways. However, no studies have investigated the potential role of E2 in combination with 5-FU chemotherapy. Exploring the role of E2 with or without 5-FU could offer a more suitable alternative and/or better-targeted therapy against CRC. Thus, this research has aimed to measure the monotherapy and dual therapy effects of 17β-estradiol (E2) and 5-Fluorouracil (5-FU) on cell cycle progression (Sub-G_1_, G_0_/G_1_, S, and G_2_/M phases), live cells, and cell death (early apoptosis, late apoptosis, and dead cells) against HT-29 female and SW480 male primary CRC cells and SW620 male metastatic CRC cells, in vitro using flow cytometry analysis.

## 2. Materials and Methods

### 2.1. Treatments and Reagents

The 17β-estradiol (E2) (E8875-5G) was obtained from Sigma-Aldrich Co. (St. Louis, MO, USA). The 5-Fluorouracil (5-FU) chemotherapy was sourced from Hospira Australia Ltd. (Melbourne, Australia). In addition, all the cell culture materials, including Dulbecco’s Modified Eagle’s Media (DMEM) (#10566032), foetal bovine serum (FBS) (#A3160802), and antibiotic-antimycotic solution (#15240062), were provided from Thermo Fisher Scientific (Waltham, MA, USA).

### 2.2. Colon Cancer Cell Lines and Culture Conditions

Human female (HT-29) and male (SW480) primary colon cancer cell lines and metastatic male (SW620) colon cancer cell lines were obtained from the American Type Culture Collection (ATCC; Manassas, VA, USA). The cells were cultured in DMEM supplemented with 10% FBS and 1% antibiotic-antimycotic solution and incubated at 37 °C and 5% CO_2_. More information about the features of these selected cells in correlation with their endogenous expression of estrogen receptors (ERα and/or ERβ) was represented in the previous paper [53].

### 2.3. Treatment Regimes

The colon cancer cells were first treated with a wide range of E2 and 5-FU concentrations for 72 h, and the IC_50_ was determined by applying the 3-(4,5- 176 Dimethylthiazol-2-yl)-2,5-Diphenyltetrazolium Bromide (MTT) cytotoxicity assay (Appendix A). In particular, the IC_50_ values of the E2 were 11.7 µM for the HT-29 cells, 12 µM for the SW480 cells, and 12.5 µM for the SW620 cells; the IC_50_ values of the 5-FU were 50.7 µM for the HT-29 cells, 54.2 µM for the SW480 cells, and 66.9 µM for the SW620 cells (Appendix A). From the MTT data, it can be noted that the colon cancer cells have an almost similar dose response in the cell viability inhibition and IC_50_ values (Appendix A).

For the flow cytometry analysis, the cells were seeded at 2 × 10^5^ per well in 6-well plates for 24 h to achieve 30%-50% confluency and then treated with the E2 at 10 nM and 5-FU at 50 µM, alone or combined for 48 h, for all the colon cancer cells. These doses and the time point were applied here to ensure that any effects of monotherapies and dual combination therapy could be precisely analysed by cell cycle and cell death, particularly the apoptosis type. The treatment groups were the untreated control (CT), the E2 and 5-FU monotherapies, and the dual combination therapy (E+F). Three independent experiments performed all treatments in triplicate (*n* = 3).

### 2.4. Cell Cycle Analysis Using PI Staining and Flow Cytometry

Following the treatment regimens, the HT-29, SW480, and SW620 cells were trypsinised and washed with PBS (500× *g* for 5 min), then fixed in ice-cold 70% ethanol for 24 h at 4 °C. After that, the cells were treated with 20 µg/mL of RNase A (#12091021; Thermo Fisher) for 15 min and then stained with 2 µg/mL propidium iodide (PI) (#P1304MP; Thermo Fisher) for 30 min in the dark at room temperature. Consequently, the stained cells were analysed by applying the Acea Novocyte 3000 flow cytometer (Agilent Technologies, Santa Clara, CA, USA). The cell numbers in the cell cycle phases: Sub-G_1_, G_0_/G_1_, S, and G_2_/M were demonstrated via the NovoExpress software cell cycle algorithm for 20,000 events. All the treatments were performed in triplicate (*n* = 3) in three independent experiments. The data represented the percentage of each cell cycle phase as the mean ± SD.

### 2.5. Cell Death Analysis Using Annexin V-FITC/PI Staining and Flow Cytometry

Cell death was assessed by an Annexin V-FITC/PI Assay Kit (#V13245; Thermo Fisher Scientific) following the kit’s protocol. Following the treatment regimens, the HT-29, SW480, and SW620 cells were collected and washed twice with ice-cold PBS and then re-suspended in 100 µL of 1× Annexin V (AV) binding buffer. After that, a mixture of AV-FITC (5 µL) and PI (1 µL) was added to each of the cell suspensions and incubated for 15 min in the dark at room temperature to allow the cells to be stained. Then, the AV binding buffer (400 µL) was added. The cells were located on the ice and directly analysed with the NovoCyte 3000 flow cytometry (Agilent Technologies, Santa Clara, CA, USA), measuring the fluorescence emissions at 530 nm.

In three independent experiments, all treatments were performed in triplicate (*n* = 3). The data represented the percentage of cells (mean ± SD) in the different cell death stages as follows: live (unstained), early apoptotic (AV+/PI−) and late apoptotic (AV+/PI+), and dead (AV−/PI+) cells. The dead cells are indicated for the very late stages of apoptosis and can be associated with secondary necrosis.

### 2.6. Statistical Analysis

The cell cycle and apoptosis were analysed by determining the percentage of cells in each phase/stage. The data were expressed as mean and standard deviation (*n* = 3). As the data were non-parametric, a Kruskal–Wallis and a Conover–Inman post-hoc test were applied using Stats Direct software (Stats Direct Ltd., Altrincham, UK) to determine the statistical significance of the data. The data were represented in a bar chart; symbols such as a green asterisk (*) were used when there was a significant increase, while the red asterisk (*) was used when there was a significant decrease. The results were considered statistically significant when *p* ≤ 0.05.

#### 2.6.1. Analysis of Monotherapies Effect

The statistical significance of the individual drugs (E2 or 5-FU) was determined by comparison with the untreated control cells (CT).

#### 2.6.2. Analysis of Dual Therapy Effect

The statistical significance of dual therapy (E+F) was determined by comparison with the CT and the monotherapies (E2 and 5-FU). The effectiveness of dual treatment on cell cycle/cell death in CRC cell lines was classified as interactive, non-interactive, or no effect, which are defined as follows:

Interactive effect: the dual therapy (E+F) was associated with a significant cell accumulation in any phase of the cell cycle/reduction in live cells/induction of cell death compared to the CT and the monotherapies (E2 and 5-FU).

Non-interactive effect: the dual therapy (E+F) was associated with a significant cell accumulation in any phase of the cell cycle/reduction in live cells/induction of cell death compared to the CT, and/or there was no significant different or it was significantly less compared to one or both treatments when applied alone.

No effect: the dual therapy (E+F) had no significant difference in cell accumulation in any phase of the cell cycle/reduction in live cells/induction of cell death compared to the CT, and/or it was significantly less when compared to the monotherapies.

## 3. Results

### 3.1. Cell Cycle Results

#### 3.1.1. The Monotherapy Effect of 7β-estradiol

The E2 monotherapy significantly increased the cells’ accumulation in the sub-G_1_ phase (Figure 1 and Table 1) and decreased the cells’ accumulation in the G_0_/G_1_ phase when compared to untreated control cells (CT) in all three of the investigated colon cancer cell lines (HT-29, SW480, and SW620), *p* ≤ 0.05 (Figure 1). A differential effect of E2 was observed in the S phase which was dependent on the type of CRC cells and which did not cause any effect in the S phase on the HT-29 female CRC cells (Figure 1). At the same time, it significantly increased the cells’ accumulation in the S phase in the SW480 male CRC cells compared to the CT, *p* ≤ 0.05 (Figure 1 and Table 1). However, it significantly decreased the cell accumulation in the S phase in the SW620 male metastatic CRC cells compared to the CT, *p* ≤ 0.05 (Figure 1 and Table 1). The G_2_/M phase was not affected by E2 monotherapy in the HT-29 and SW480 cells, while the cells were significantly reduced in this phase in the SW620 metastatic cells compared to the CT, *p* ≤ 0.05 (Figure 1).

#### 3.1.2. The Monotherapy Effect of 5-Fluorouracil

The monotherapy of 5-FU significantly increased the accumulation of the cells in the sub-G_1_ phase and decreased the cell accumulation in the G_0_/G_1_ and G_2_/M phases when compared to CT in all the investigated CRC cell lines (HT-29, SW480, and SW620), *p* ≤ 0.05 (Figure 1 and Table 1). In addition, the cells were also significantly accumulated at the S phase in the HT-29 female and SW480 male primary CRC cells compared to the CT, *p* ≤ 0.05 (Figure 1 and Table 1); meanwhile, no effect was seen in the S phase for the SW620 male metastatic CRC cells compared to the CT, *p* ≤ 0.05 (Figure 1 and Table 1).

#### 3.1.3. The Dual Therapy Effect of 7β-estradiol and 5-Fluorouracil

In both the HT-29 female and the SW480 male primary CRC cells, the dual therapy (E+F) was shown to have an interactive effect, causing the significant accumulation of cells within the sub-G_1_ phase of the cell cycle compared to the CT and monotherapies (E2 and 5-FU) (*p* ≤ 0.05) (Figure 1 and Table 1). In the SW620 male metastatic CRC cells, the dual therapy was associated with a non-interactive effect in the sub-G_1_ phase; there was a significant accumulation of cells in this phase compared to the CT and one of the monotherapies (in particular to 5-FU only) (*p* ≤ 0.05), with an almost similar effect to that of the E2 monotherapy (Figure 1 and Table 1).

At the same time, in all the investigated CRC cells (HT-29, SW480, and SW620), the dual therapy was shown to have a significant reduction in cell accumulation in the G_0_/G_1_ phase compared to the CT and monotherapies (*p* ≤ 0.05) (Figure 1). Regarding the S phase, the E+F therapy caused a non-interactive effect in this phase which linked with a significant accumulation of cells in the HT-29 and SW480 cells; meanwhile, it caused a significant reduction in SW620 cell accumulation compared to one of the monotherapies and CT (*p* ≤ 0.05) (Figure 1). Moreover, the dual therapy did not cause any effect in the G_2_/M phase in the HT-29 cells, while there was a significant reduction in cell accumulation at this phase in the SW480 and SW620 cells compared to the CT and monotherapies (*p* ≤ 0.05) (Figure 1).

The significant arrest of the cell cycle phases by either monotherapies (E2 or 5-FU) or dual therapy (E+F) in human colon cancer cell lines is summarised and represented in Table 1, with their *p* values

### 3.2. Cell Death Results

#### 3.2.1. The Monotherapy Effect of 7β-estradiol

In all the investigated CRC cell lines (HT-29, SW480, and SW620), the E2 monotherapy significantly reduced the live cells and induced apoptosis; in particular, there was the significant induction of late apoptosis seen in the HT-29 cells, the early apoptosis observed in the SW480 cells, and both the early and the late apoptosis in the SW620 cells when compared to the CT, *p* ≤ 0.05 (Figure 2, Table 2 and Table 3). No significant induction was seen on the dead cells in all the cell lines compared to the CT (Figure 2).

#### 3.2.2. The Monotherapy Effect of 5-Fluorouracil

In all the investigated colon cancer cells (HT-29, SW480, and SW620), the 5-FU monotherapy significantly reduced the live cells and induced early apoptosis in the HT-29 and SW620 cells and both early and late apoptosis in the SW480 cells compared to the CT, *p* ≤ 0.05 (Figure 2, Table 2 and Table 3). No significant induction was observed on the dead cells in the HT-29 and SW480 primary CRC cells, whilst significant induction was seen in the SW620 metastatic CRC cells compared to the CT, *p* ≤ 0.05 (Figure 2).

#### 3.2.3. The Dual Therapy Effect of 7β-estradiol and 5-Fluorouracil

In the HT-29 cells, the dual therapy (E+F) was shown to have an interactive effect, causing a significant reduction in live cells when compared to the CT and monotherapies (*p* ≤ 0.05) (Figure 2 and Table 2), while in the remaining CRC cells (SW480 and SW620) the dual therapy was revealed to have had a non-interactive effect, which was associated with a significant reduction in live cells but, compared to the CT, with a significant differential effect against the monotherapies (*p* ≤ 0.05) (Figure 2 and Table 2).

Interestingly, the treated HT-29 and SW480 primary CRC cells with the dual therapy were associated with an interactive effect on apoptosis induction, in which there was a significant induction of early or late apoptotic cells, respectively, compared to the CT and monotherapies (*p* ≤ 0.05) (Figure 2, Table 2 and Table 3). However, the dual therapy did not cause any interactive effects on cell death induction in the SW620 metastatic CRC cells, which was associated with a differential effect against the monotherapies; meanwhile, there was a significant induction of early and late apoptosis compared to the CT, (*p* ≤ 0.05) (Figure 2 and Table 3).

The significant reduction in live cells and the induction of cell death by the monotherapies (E2 or 5-FU) and dual therapy (E+F) in human colon cancer cells are summarised and represented in Table 2 and Table 3.

## 4. Discussion

The current study explored the anti-cancer activity of 17β-oestradiol (E2) as monotherapy and dual therapy with 5-Fluorouracil (5-FU) in vitro against the female (HT-29) and male (SW480) primary CRC cells, as well as the male (SW620) metastatic CRC cells. This is the first study to explore the in vitro anti-cancer activity of combining E2 with 5-FU against CRC cells. The findings here showed that the E2 and 5-FU monotherapies significantly promoted the cell cycle arrest and apoptosis induction in all the investigated CRC cell lines (HT-29, SW480, and SW620), suggesting that E2 monotherapy could be a promising therapeutic strategy for both primary and metastatic CRC types. However, the greatest anti-cancer actions were observed with dual therapy (E+F) against the female and male primary CRC cells (HT-29 and SW480), suggesting that E2 could enhance the activity of 5-FU chemotherapy and could be used as an alternative treatment combination strategy for the early stage of CRC, for both the female and the male genders.

Notably, the monotherapy of 5-FU (50 µM) at 48 h caused a significant increase in the accumulation of the cells in the sub-G_1_ and S phases in the HT-29 female and SW480 male primary CRC cells and in the sub-G_1_ phase only in the SW620 male metastatic CRC cells, with a significant reduction in live cell numbers and the induction of early apoptosis in all three of the investigated CRC cells when compared to the CT. In addition, 5-FU monotherapy induced late apoptotic cells in the SW480 cells. 5-FU produces its anti-cancer effects by suppressing thymidylate synthase (TS) and disrupting the intracellular deoxynucleotide pools needed for DNA replication [10]. Therefore, it can inhibit cellular proliferation by controlling the critical molecular pathway molecules that inhibit the DNA synthesis in the S phase and induce cell cycle arrest (G_0_/G_1_ or S phases), causing DNA damage [54]. Accordingly, 5-FU is thought to suppress cancer cell growth by inducing apoptosis and cell death [54]. Moreover, the cell cycle arrest induced by 5-FU involves the dysregulation of the critical intracellular signalling pathways and their related molecules, including activating cyclin-dependent kinase inhibitors (p21^Waf1/Cip1^ and p27^Kip1^), inhibiting the markers of proliferation, such as proliferating cell nuclear antigen (PCNA) and survivin, and upregulating the expression of the activity of the apoptosis-related proteins, such as p53 and caspase-3 [55,56,57].

The data here also demonstrated that E2 monotherapy (10 nM) at 48 h effectively arrested the cell cycle at the sub-G_1_ phase, reduced the live cells, and induced early and/or late apoptosis in all the investigated CRC cell lines (HT-29, SW480, and SW620) that either derived from the primary or metastatic tumors. Accordingly, E2 monotherapy could be highly effective as an anti-cancer agent for treating both the early and advanced stages of colon cancer in both genders. Based on the significant values in Table 1, Table 2 and Table 3, E2 monotherapy greatly arrested the cells in the sub-G_1_ phase and induced both early and late apoptosis in metastatic CRC cells compared to its effect on primary CRC cells.

Taken together, the E2 and 5-FU monotherapies had an almost similar effect on the cell cycle by arresting mainly the sub-G_1_ phase (Table 1), reducing the living cells (Table 2), and inducing early and/or late apoptosis (Table 3) in all the investigated cell lines (HT-29, SW480, and SW620). Meanwhile, it is important to mention that there is some differential sensitivity in response to monotherapies on the cell cycle and apoptosis; this is normal and is due to the diversities in the cellular and molecular features of each of the CRC cell lines applied [53], which could have induced some different anticancer responses.

In agreement with these results, earlier studies found that DLD-1 primary male CRC cells, when treated with a single therapy of E2 at a 10 nM dose for 24, 30, and/or 48 hrs, caused an inhibition of the cell proliferation [58,59,60], the arrest the cell cycle in the sub-G_1_ phase [59], and apoptosis induction [59,60,61]. Mechanistically, the E2 anticancer activity has occurred through activating the caspase-3 induction and increasing the poly ADP-ribose polymerase (PARP) cleavage [59,60,61]. In addition, most of the current studies have reported that the actions of E2 in CRC are receptor-dependent and have also shown that ERα is oncogenic, whereas ERβ exerts anti-cancer activities [44,62,63,64], which, consequently, can activate the p38 mitogen-activated protein kinase (p38/MAPK) pathway [65]. Several human CRC cell lines are endogenously overexpressed with ERβ receptors. For example, the HT-29, SW480, and DLD-1 primary cells were reported to express ERβ and lack ERα [44,58,61,63]. In contrast, the SW620 metastatic cells express ERα and ERβ [64].

Interestingly, additional studies found that the ERβ re-expression exogenously, along with E2 treatment at 10 nM for 24 h, could be a very effective therapeutic strategy to produce anti-cancer activity for the primary colon cancer cells (SW480 and HT-29) [35,66,67]. They showed that this treatment strategy caused cell cycle arrest at the G_1_ phase in female (HT-29) and male (SW480) colon cancer cells [35,66] by decreasing the expression of MYC [67,68], MYB, and the PROX1 [67] oncogenes and increasing the expression of the p21^Waf1/Cip1^ and p27^Kip1^ cell cycle inhibitors [35]. On the other hand, Edvardsson et al. (2013) [67] reported that E2 treatment following ERβ activation in the SW620 metastatic CRC cells showed no effects on the MYC expression. The metastatic CRC cells could respond to this treatment strategy if investigated for a long treatment period, such as 48 h rather than 24 h, as the E2 monotherapy here showed significant anti-cancer activity in this type of cells at 48 h.

Remarkably, the data here also revealed a significant accumulation of the cells at the sub-G_1_ phase and the induction of apoptosis in HT-29 female and SW480 male primary CRC cells when treated with dual combination therapy (E+F) compared to the action of the monotherapy (E2 and 5-FU) and the CT. Meanwhile, dual treatment has a lower effect in SW620 male metastatic CRC cells. These findings suggested that E2 can enhance the activity of 5-FU, and this combination regimen could be a promising therapeutic approach for the early stages of CRC for both genders. However, a lower concentration of E2 and 5-FU may be needed for a dual therapy strategy in treating the metastatic SW620 CRC cells to produce an interactive effect on the cell cycle progression and apoptosis because, when the percentage of cells in the sub-G_1_ phase is considered, the E2 monotherapy has caused maximum efficacy on the cell cycle in which 70% of the cells in the sub-G_1_ phase had accumulated; thus, using a lower dose, particularly for E2 when combined with 5-FU, could produce an interactive effect in SW620 cells.

The findings from the flow cytometric analysis in the current study demonstrated that all the treated CRC cells were arrested at the sub-G1 phase associated with apoptosis induction. The detection of most of the treated cells in the sub-G1 phase demonstrated strong evidence of apoptosis induction [69]. Apoptosis is subsequently linked with DNA damage induction, which leads to the arrest of the cell cycle progression [26]. The cell cycle includes a series of events within a cell divided into interphase (G_1_, S, and G_2_ phases) and mitosis (M phase), which leads to its division and duplication [3,4]. The G_1_ phase is a crucial part of the cell cycle as it provides the signal to the cell to permit it to enter the cell division stage [69]. Thus, regulating the cell cycle progression and apoptosis is a practical approach to controlling tumor growth [69].

This first study demonstrated that 17β-estradiol could enhance the anti-cancer activities of 5-Fluorouracil chemotherapy in treating the human female and male colon cancer cells; this was preferentially provoked in the sub-G, phase associated with the induction of apoptosis. Nevertheless, further studies are still needed to explore the genomic and non-genomic effects of E2 and 5-FU dual therapy for CRC to fully elucidate the mechanisms underlying their anti-cancer actions in order to corroborate the observations here. Future studies should also investigate their anti-cancer activities in female and male experimental models (*in vivo*) to determine their precise therapeutic values in CRC.

The17β-estradiol can initiate its cellular signals by triggering and activating its nuclear receptors (ERα and ERβ), which mediate major genomic actions [41,42,43]. It is important to mention that female sex steroid can produce pro- or antitumorigenic effects, based on the origin of the cancer type, the expression pattern of the estrogen endogenous receptors in the tissue, and the malignancy stage [70,71]. In this context and in contrast to CRC, estrogens (either endogenous or exogenous) have been revealed to promote cell proliferation and tumor progression in the breast and uterus [72]. These pro-tumorigenic actions are mediated via the ERα, which is predominant in these tissues [72]. However, the activities of E2 in CRC are receptor-dependent; ERα produces oncogenic actions, while ERβ produces anticancer actions [64]. Thus, it is essential to consider the cancer types when using the therapeutic strategy, including the female sex steroid hormones.

## 5. Conclusions

In conclusion, the most responsive CRC cells to the dual therapy (E+F) regimen were the primary CRC cells (HT-29 and SW480), while the metastatic CRC cells (SW620) were the least responsive to this treatment strategy. Meanwhile, the SW620 cells were the most responsive CRC cell type to the E2 monotherapy compared to the primary CRC cells. The investigated E2 treatment strategies consistently arrested the cell cycle progression at the sub-G1 phase in conjunction with early and/or late apoptosis induction in CRC cells. These results suggested that E2 monotherapy could be more effective for the advanced stages of CRC and that the E+F dual therapy could be an alternative potential combination regimen for the early stages of CRC, and, interestingly, it could be for both genders. Meanwhile, additional studies are still required to explore those promising treatment strategies at the molecular levels, *in vitro* and *in vivo*.

## Figures and Tables

**Figure 1 medsci-10-00062-f001:**
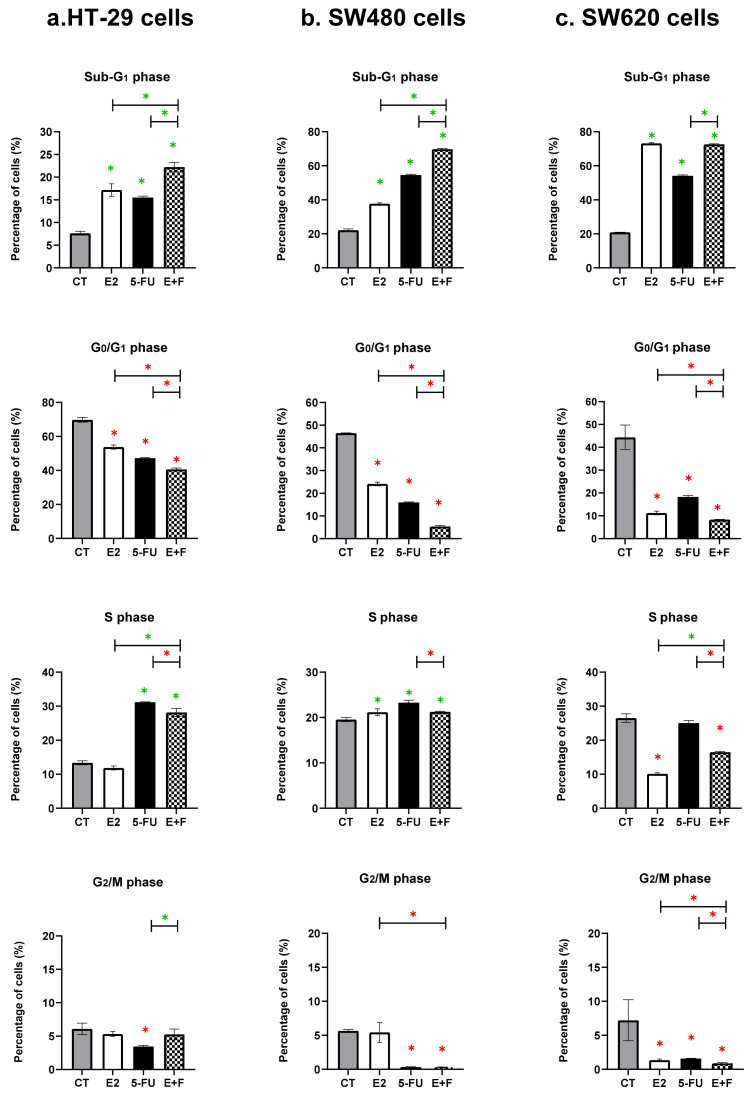
Percentage of cells (mean ± SD of three independent experiments in triplicate) in the cell cycle phases (sub-G_1_, G_0_/G1, S, G_2_/M) in untreated control cells (CT) and following treatments with 7β-estradiol (E2) at 10 nM and 5-Fluorouracil (5-FU) at 50 µM, alone and in combination (E+F), for 48 h in the: (**a**) HT-29 female and (**b**) SW480 male primary colon cancer cells and in the (**c**) SW620 male metastatic colon cancer cells, using the PI staining and flow cytometry analysis. Data were represented in the bar chart; symbols such as the green asterisk (*) were used when there was a significant increase, while the red asterisk (*) was used when there was a significant decrease. Results were considered statistically significant when *p* ≤ 0.05; monotherapies were statistically compared to CT only, while dual therapy was statistically compared to CT and monotherapies (Section 2.6).

**Figure 2 medsci-10-00062-f002:**
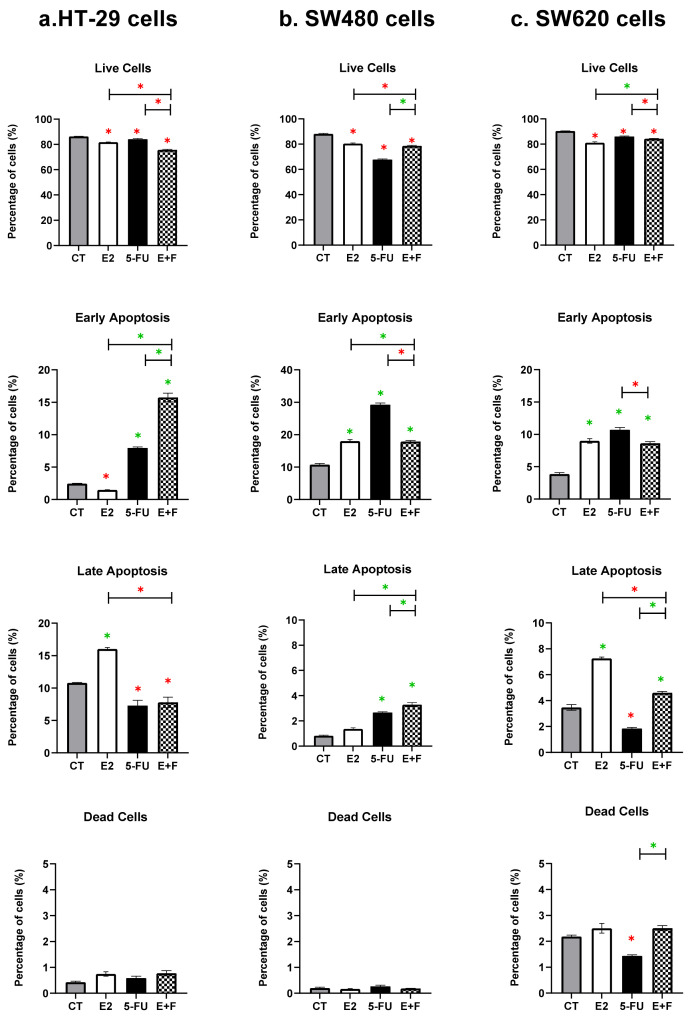
Percentage of the live cells and early and late apoptotic cells alongside dead cells (mean ± SD of three independent experiments in triplicate) in untreated control cells (CT) and following treatments with 7β-estradiol (E2) at 10 nM and 5-Fluorouracil (5-FU) at 50 µM at µM alone and in combination (E+F) for 48 in the: (**a**) HT-29 female and (**b**) SW480 male primary colon cancer cells, and in the (**c**) SW620 male metastatic colon cancer cells, using the AV/PI staining and flow cytometry analysis. Data were represented in the bar chart; symbols such as the green asterisk (*) were used when there was a significant increase, while the red asterisk (*) was used when there was a significant decrease. Results were considered statistically significant when *p* ≤ 0.05; monotherapies were statistically compared to CT only, while dual therapy was statistically compared to the monotherapies and CT (Section 2.6).

**Table 1 medsci-10-00062-t001:** The summary of significant arrested cell cycle phases by monotherapies and dual therapy in human colon cancer cells (*p* ≤ 0.05).

Colon Cancer Cell Lines	Monotherapies Compared to CT Only		Dual Therapy Compared to CT & Monotherapies
	E2	5-FU	E+F
HT-29 cells	Sub-G_1_ phase *Significant arrest* *vs. CT, p = 0.0007*	Sub-G_1_ phase *Significant arrest* *vs. CT, p = 0.0133* S phase *Significant arrest* *vs. CT, p < 0.0001*	Interactive effect at Sub-G_1_ phase *Significant arrest vs. CT, p < 0.0001* *Significant arrest vs. E2, p = 0.0133* *Significant arrest vs. 5-FU, p = 0.0007*
SW480 cells	Sub-G_1_ phase *Significant arrest* *vs. CT, p = 0.0063* S phase *Significant arrest* *vs. CT, p = 0.0166*	Sub-G_1_ phase *Significant arrest* *vs. CT, p = 0.0002* S phase *Significant arrest* *vs. CT, p = 0.0001*	Interactive effect at Sub-G_1_ phase *Significant arrest vs. CT, p < 0.0001* *Significant arrest vs. E2, p < 0.0001* *Significant arrest vs. 5-FU, p = 0.0063*
SW620 cells	Sub-G_1_ phase *Significant arrest* *vs. CT, p < 0.0001*	Sub-G_1_ phase *Significant arrest* *vs. CT, p = 0.0063*	Non-interactive effect at Sub-G_1_ phase *Significant arrest vs. CT, p < 0.0001* *No significant arrest vs. E2, p = 0.2017* *Significant arrest vs. 5-FU, p = 0.0063*

The summary of the significant arrest of cell cycle phases by monotherapies: 7β-estradiol (E2) at 10 nM and 5-Fluorouracil (5-FU) at 50 µM and dual therapy (E+F) at 48 h in the HT-29 female and SW480 male primary CRC cells and in the SW620 male metastatic CRC cells. The treatment effects were statistically determined based on the data represented in Figure 1 using the statistical analysis methods of Section 2.6.

**Table 2 medsci-10-00062-t002:** The summary of the significant reduction in living cells by monotherapies.

Colon Cancer Cell Lines	Monotherapies Compared to CT Only		Dual Therapy Compared to CT & Monotherapies
	E2	5-FU	E+F
HT-29 cells	Significant reduction *vs. CT, p < 0.0001*	Significant reduction *vs. CT, p = 0.0063*	Interactive effect on live cells *Significant reduction vs. CT, p = 0.0063* *Significant reduction vs. E2, p = 0.0063* *Significant reduction vs. 5-FU, p < 0.0001*
SW480 cells	Significant reduction *vs. CT, p = 0.0063*	Significant reduction *vs. CT, p < 0.0001*	Non-interactive effect on live cells *Significant reduction vs. CT, p < 0.0001* *Significant reduction vs. E2, p = 0.0063* *Significant increase vs. 5-FU, p = 0.0063*
SW620 cells	Significant reduction *vs. CT, p < 0.0001*	Significant reduction vs. CT, *p* = 0.0063	Non-interactive effect on live cells *Significant reduction vs. CT, p < 0.0001* *Significant increase vs. E2, p = 0.0063* *Significant reduction vs. 5-FU, p = 0.0052*

7β-estradiol (E2) at 10 nM and 5-Fluorouracil (5-FU) at 50 µM and dual therapy (E+F) at 48 h in the HT-29 female and SW480 male primary CRC cells and in the SW620 male metastatic CRC cells. The treatment effects were statistically determined based on the data represented in Figure 2, using the statistical analysis methods of Section 2.6.

**Table 3 medsci-10-00062-t003:** The summary of significant induction of early/late apoptosis by monotherapies and dual therapy in human colon cancer cell lines, (*p* ≤ 0.05).

Colon Cancer Cell Lines	Monotherapies Compared to CT Only		Dual Therapy Compared to CT & Monotherapies
	E2	5-FU	E+F
HT-29 cells	Late apoptosis Significant induction *vs. CT, p = 0.0372*	Early apoptosis Significant induction *vs. CT, p = 0.0063*	Interactive effect on early apoptosis *Significant induction vs. CT, p < 0.0001* *Significant induction vs. E2, p < 0.0001* *Significant induction vs. 5-FU, p = 0.0063*
SW480 cells	Early apoptosis Significant induction *vs. CT, p = 0.0047*	Early apoptosis Significant induction *vs. CT, p = 0.0001* Late apoptosis Significant induction *vs. CT, p < 0.0001*	Interactive effect on late apoptosis *Significant induction vs. CT, p < 0.0001* *Significant induction vs. E2, p < 0.0001* *Significant induction vs. 5-FU, p = 0.0064*
SW620 cells	Early apoptosis Significant induction *vs. CT, p = 0.0007* Late apoptosis Significant induction *vs. CT, p < 0.0001*	Early apoptosis Significant induction *vs. CT, p < 0.0001*	Non-interactive effect on early apoptosis *Significant induction vs. CT only, p = 0.0133* *No significant induction vs. E2, p = 0.0578* *Significant reduction vs. 5-FU, p = 0.0007* Non-interactive effect on late apoptosis *Significant induction vs. CT, p = 0.0063* *Significant induction vs. E2, p = 0.0063* *Significant reduction vs. 5-FU, p < 0.0001*

The summary of the significant induction of cell death (early apoptosis, late apoptosis, and/or dead cells) by monotherapies: 7β-estradiol (E2) at 10 nM and 5-Fluorouracil (5-FU) at 50 µM and dual therapy (E+F) at 48 h in the HT-29 female and SW480 male primary CRC cells and in the SW620 male metastatic CRC cells. The treatment effects were statistically determined based on the data represented in Figure 2, using the statistical analysis methods of Section 2.6.

## Data Availability

Not applicable.

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
