# Peer review of "17β-estradiol Enhances 5-Fluorouracil Anti-Cancer Activities in Colon Cancer Cell Lines"

_medsci, 2022, doi:10.3390/medsci10040062_

Round 1
Reviewer 1 Report
Dear author,
The author proposed that a dual therapy of sex hormone (17β-estradiol, E2) and 5-Fluorouracil against early colorectal cancer along with several evidence (cell cycle analysis and apoptosis assay). The manuscript is well-written and fits the scope of this journal. However, there are some points should be explained and edited. Details are followed.
1. Line 113-116: “the IC50 concentrations of E2 at 10 nM and 5-FU at 50 mM alone or combined for 48h and these concentrations were previously determined by the 3-(4,5-176 Dimethylthiazol-2-yl)-2,5-Diphenyltetrazolium Bromide (MTT) cytotoxicity assay at 72h (data not shown). Both E2 and 5-FU showed the same IC50 for all three CRCs (HT-29, SW480, and SW620 cells)? Based on cell cycle, apoptosis assay in this study, they don’t seem have a same or similar sensitivity to each therapy. Please explain about it.
2. Figure 1c and Table 1: Based on Figure 1c, E2 (10 nM) monotherapy already dramatically increased Sub-G1 phase population (about 70%) in SW620 and it was not further increased by dual therapy. It was interpretated by “None-interactive effect at Sub-G1 phase” in SW620 cells at Table 1. However, when the percentage of cells (%) in Sub-G1 phase was considered, SW620 cells were with E2 monotherapy group maybe maximum efficacy by itself. Thus, it may not have a room for “interactive effect”. To rule out this kind of claim, lower concentrations of E2 with 5-FU will give us to make it clear.
3. Line 94, Line 397: ‘in vitro’ and ‘in vivo’ should be italics
Author Response
The author proposed that a dual therapy of sex hormone (17β-estradiol, E2) and 5-Fluorouracil against early colorectal cancer along with several evidence (cell cycle analysis and apoptosis assay). The manuscript is well-written and fits the scope of this journal.
Answer: Thank you very much for your kind comment.
However, there are some points should be explained and edited. Details are followed.
- Line 113-116: “the IC concentrations of E2 at 10 nM and 5- FU at 50 mM alone or combined for 48h and these concentrations were previously determined by the 3-(4,5-176 Dimethylthiazol-2-yl)-2,5-Diphenyltetrazolium Bromide (MTT) cytotoxicity assay at 72h (data not shown). Both E2 and 5-FU showed the same IC for all three CRCs (HT-29, SW480, and SW620 cells)? Based on cell cycle, apoptosis assay in this study, they don’t seem have a same or similar sensitivity to each therapy. Please explain about it.
Answer: Thank you for your comment. I have added more clarifications in the Treatment regimens (section 2.3) about the exact IC50 values for each cell supported with Supplementary Figure 1 in the 2.3 section. In fact, also as u can see in the results of E2 and 5-FU monotherapies have almost a similar effect on the cell cycle by arresting mainly the sub-G1 phase (Table 1), reducing living cells (Table 2) and inducing early and/or late apoptosis (Table 3) in all the investigated cell lines (HT-29, SW480, and SW620).
Meanwhile, some differential sensitivity in response to monotherapies on cell cycle and apoptosis it is normal due to the diversities in the cellular and molecular features of each of the CRC cell lines applied [53], which could have normally induced some different anticancer responses. This information has been added in the discussion section (paragraph no 4) for clarification.
- Figure 1c and Table 1: Based on Figure 1c, E2 (10 nM) monotherapy already dramatically increased Sub-G1 phase population (about 70%) in SW620 and it was not further increased by dual therapy. It was interpretated by “None-interactive effect at Sub-G1 phase” in SW620 cells at Table 1. However, when the percentage of cells (%) in Sub-G1 phase was considered, SW620 cells were with E2 monotherapy group maybe maximum efficacy by itself. Thus, it may not have a room for “interactive effect”. To rule out this kind of claim, lower concentrations of E2 with 5-FU will give us to make it clear.
Answer: Thank you for your important comment. I have added this point in the discussion section (paragraph no 6) for more clarification.
- Line 94, Line 397: ‘in vitro’ and ‘in vivo’ should be italics
Answer: Thank you for your comment. I have modified the ‘in vitro’ and ‘in vivo’ to the italic format in mentioned lines and checked also throughout the texts.
I hope these responses and changes made are satisfactory and you will consider the article suitable for publication in your journal. Please do not hesitate to contact me should you require any further details.

Reviewer 2 Report
It was appropriately constructed for the complex (17β-estradiol and 5-Fluorouracil) apoptosis effect in chemotherapy. There are a few corrections before publication.
1.It seems that this paper should have a dose-dependent cytotoxicity assay to support the content (as a supporting data?) (line 116, data not shown?).
2.I understand the author's intentions, but the title need not refer to both female and male at the same time.
3.Some cells (HT-29 and SW480,,,) are not representative of males and females. Throughout the text, it is better to focus on the specific cells.
4.I don't think it's necessary to mention about cyclin D1, BCL2 (cytochrome C and caspase-3 in the intro part(line 34-38). It would be better to describe other content related to the text.
Author Response
It was appropriately constructed for the complex (17β-estradiol and 5-Fluorouracil) apoptosis effect in chemotherapy.
Answer: Thank you very much for your kind comment.
There are a few corrections before publication.
- 1. It seems that this paper should have a dose-dependent cytotoxicity assay to support the content (as a supporting data?) (line 116, data not shown?).
Answer: Thank you for your comment. I have added the MTT results as supplementary data (Supplementary Figure 1), and you can find that data kindly on the final page.
- 2. I understand the author's intentions, but the title need not refer to both female and male at the same time.
Answer: Thank you for your comment. I updated the title to “17β-estradiol enhances 5-Fluorouracil anti-cancer activities in colon cancer cell lines".
- Some cells (HT-29 and SW480,,,) are not representative of males and females. Throughout the text, it is better to focus on the specific cells.
Answer: Thank you for your comment. In fact, HT-29 cells were derived from a female patient, and the SW480 cells were derived from a male patient. So, the HT-29 and SW480 are representative cells from female and male patients, respectively. Indeed, it is important to mention that to demonstrate the effect of the investigated treatment strategies between the female and male cells; and precisely observed also if the male cells have an anti-cancer response to E2 either alone or in combination with 5-FU. In fact, determining the gender derivative cells is important for this research. I already also mentioned in section 2.2 of Methods that more information about the features of these selected cells in correlation to their endogenous expression of estrogen receptors (ERα and/or ERβ) has been represented in the previous paper [53].
- I don't think it's necessary to mention about cyclin D1, BCL2 (cytochrome C and caspase-3 in the intro part (line 34-38). It would be better to describe other content related to the text.
Answer: Thank you for your comments. I have removed unnecessary information from that sentence in paragraph no 1 of the introduction.

Reviewer 3 Report
I found the manuscript "17β-estradiol enhances 5-Fluorouracil anti-cancer activities in 2 female and male colon cancer cell lines" very interesting, easy to read and well-written. I believe that it will be a good addition to the literature. The obtained results constitute a step toward new strategies for colon cancer treatment.
I kindly suggest reviewing the manuscript to improve the language (verb tenses etc., e.g. lines: 179, 184… ). The device Novocyte 3000 is described twice (lines 123-124 and 129-130).
Author Response
I found the manuscript "17β-estradiol enhances 5-Fluorouracil anti-cancer activities in 2 female and male colon cancer cell lines" very interesting, easy to read and well-written. I believe that it will be a good addition to the literature. The obtained results constitute a step toward new strategies for colon cancer treatment.
Answer: Thank you very much for your kind comments, and I am glad you enjoyed my paper.
- I kindly suggest reviewing the manuscript to improve the language (verb tenses etc., e.g. lines: 179, 184…).
Answer: Thank you for your comment. Apologies for the mistakes. I have re-corrected the verb tenses in mentioned lines and checked also throughout the texts.
- The device Novocyte 3000 is described twice (lines 123-124 and 129-130).
Answer: Thank you for your comment. I have removed the repetition in section 2.4.

Round 2
Reviewer 1 Report
Dear author,
I am really satisfied with the revised version of manuscript.
Reviewer 2 Report
Accept in present form